# Cell Cultures as a Versatile Tool in the Research and Treatment of Autoimmune Connective Tissue Diseases

**DOI:** 10.3390/cells12202489

**Published:** 2023-10-19

**Authors:** Adam Ejma-Multański, Anna Wajda, Agnieszka Paradowska-Gorycka

**Affiliations:** Department of Molecular Biology, National Institute of Geriatrics, Rheumatology and Rehabilitation, 02-637 Warsaw, Poland; anna.wajda@spartanska.pl (A.W.); agnieszka.paradowska-gorycka@spartanska.pl (A.P.-G.)

**Keywords:** ACTD, cell cultures, systemic sclerosis, lupus erythematosus, rheumatoid arthritis, primary cell cultures, 2D cultures, 3D cultures, PBMC, fibroblasts

## Abstract

Cell cultures are an important part of the research and treatment of autoimmune connective tissue diseases. By culturing the various cell types involved in ACTDs, researchers are able to broaden the knowledge about these diseases that, in the near future, may lead to finding cures. Fibroblast cultures and chondrocyte cultures allow scientists to study the behavior, physiology and intracellular interactions of these cells. This helps in understanding the underlying mechanisms of ACTDs, including inflammation, immune dysregulation and tissue damage. Through the analysis of gene expression patterns, surface proteins and cytokine profiles in peripheral blood mononuclear cell cultures and endothelial cell cultures researchers can identify potential biomarkers that can help in diagnosing, monitoring disease activity and predicting patient’s response to treatment. Moreover, cell culturing of mesenchymal stem cells and skin modelling in ACTD research and treatment help to evaluate the effects of potential drugs or therapeutics on specific cell types relevant to the disease. Culturing cells in 3D allows us to assess safety, efficacy and the mechanisms of action, thereby aiding in the screening of potential drug candidates and the development of novel therapies. Nowadays, personalized medicine is increasingly mentioned as a future way of dealing with complex diseases such as ACTD. By culturing cells from individual patients and studying patient-specific cells, researchers can gain insights into the unique characteristics of the patient’s disease, identify personalized treatment targets, and develop tailored therapeutic strategies for better outcomes. Cell culturing can help in the evaluation of the effects of these therapies on patient-specific cell populations, as well as in predicting overall treatment response. By analyzing changes in response or behavior of patient-derived cells to a treatment, researchers can assess the response effectiveness to specific therapies, thus enabling more informed treatment decisions. This literature review was created as a form of guidance for researchers and clinicians, and it was written with the use of the NCBI database.

## 1. Introduction

Autoimmune connective tissue diseases (ACTDs) are a group of complicated multisystem disorders in which patient’s immune system cells attack tissues such as joints, tendons, skeletal muscles, neural connections, cartilage, bones and other connective tissues [1]. Even though these diseases differ from each other in their symptoms as well as in affected cells, they all share the same pathological mechanism, which manifests as a prolonged state of inflammation in the affected area or areas, leading to the degeneration and degradation of afflicted structures [2]. The etiology of ACTDs remains mostly undiscovered with suspected involvement of genetic, hormonal and environmental factors [3]. Because of the unknown etiology and ACTDs’ multisystemic nature, proper diagnosis and treatment is a major challenge for healthcare systems. Therefore, cell cultures are one of the tools used to understand the molecular basis/mechanisms of/in ACTDs. Due to the lack of uniformed diagnosis criteria and the amount of ACTDs, it would be impossible to describe the cell culture types used in the research on all of them in one paper. So in this review, the authors have focused on few selected ACTDs, and the culture types used in research on them. These diseases are presented in Table 1.

Cell culture development began in the early 1900s; however, their true potential was discovered and advanced in late 1940s when they were used to grow polioviruses in the production of the polio vaccine [13]. Conducting an experiment with cell cultures allows scientists to study the in vitro morphology, physiology and genetics of a single or few types of cells of interest. While cell cultures are a powerful tool in research, it is important to remember their limitations and requirements. First of all, working with tissue cultures requires strict conditions, such as CO_2_ levels, temperature, etc. Secondly, in order to grow cells must be placed in dedicated media which maintain their physiological state and functionality [14]. The media which are used in the cell cultures described in this paper are listed in Table 2. Depending on the type of cell culture, its development might be a long and laborious process, which is why researchers often buy commercially prepared cell lines to hasten the experiment.

Connective tissue is a type of tissue that is present in every physiological system in the human body. It acts as a scaffolding for other tissues and organs, supports their functions, protects them against harmful factors and helps to repair damaged tissues [19]. During the course of ACTD-afflicted connective tissue with ongoing inflammation and, very often, fibrosis, is unable to perform its functions, thus starting a domino effect leading to deterioration of other tissues and organs. This paper aims to present a synthetic review of the literature about the employment of cell cultures in the research and treatment of ACTDs.

## 2. Fibroblast Cultures

Fibroblast cell cultures are the most basic and best established of all known culture types; therefore, they can be considered a model culture. These cells are found in almost all of the tissues of the body, including skin, bones, muscles and organs. Fibroblasts play a crucial role in maintaining the structural integrity of tissues and organs. [20] Fibroblasts produce and secrete the extracellular matrix (ECM), a complex mixture of collagen, elastin and other proteins and carbohydrates, that forms the structural framework of tissues, providing them with necessary strength and flexibility. Fibroblasts also play a key role in early stages of wound healing and immune response by inducing receptor-linked chemokine synthesis [20]. As a fundamental type of connective tissue cell, fibroblasts are more or less affected in most of the ACTDs; however, in the course of SSc [11], DM [8] and PM [8] they play major roles. Briefly, in SSc, fibroblasts are targeted by immune cells which secrete profibrotic cytokines. These cytokines disrupt the balance of ECM production and degradation, causing fibroblasts to become dysfunctional and overproduce collagen and other proteins that form scar tissue. This excess collagen causes thickening and hardening of the skin, as well as damaging the blood vessels, muscles and internal organs [21]. In DM and PM, the muscle tissue is invaded by T and B lymphocytes, similarly to SSc. But, instead of secreting profibrotic cytokines, they produce cytokines and antibodies that destroy myocytes and damage the tissue leading to muscle weakness and atrophy. Even though fibroblasts are not directly targeted in myositis, they may be damaged by infiltrating immune cells, releasing MDA-5 antigen and mobilizing anti-MDA-5 antibodies, causing an increase in inflammation and a characteristic skin rash [22].

Establishing a primary cell culture of fibroblasts involves isolating these cells from tissue of interest and culturing them in a suitable growth medium. These tissues can be easily obtained through a biopsy or discarded surgical material. After acquisition, the collected material must be dissociated into smaller pieces, whether by using a sterile scalpel and/or by enzymatic digestion to release individual cells [20]. Isolated cells should be rinsed with PBS, resuspended in DMEM with 10% fetal bovine serum (FBS) and 1% antibiotic-antimycotic addition, and maintained in 5% CO_2_ environment at 37 °C [23,24,25].

One use of fibroblast cultures in the aforementioned diseases is to study the underlying mechanisms of the pathological processes involved in their development and progression [4,26,27]. Chadli et al. conducted an experiment in which they used skin biopsies collected from patients diagnosed with early SSc to generate fibroblast cell culture. Biopsies were minced into small fragments, treated with dispase II to separate dermis from epidermis and then the dermis was treated with mixture of dispase II and collagenase II to free fibroblasts, which were cultured in standard conditions in DMEM [26]. Subsequently, RNA was extracted from the cultured cells and micro-array was performed to measure the expression levels. The researchers found that fibroblasts derived from SSc skin biopsies retained most of the diseases’ molecular phenotype, therefore highlighting the value of phenotyped fibroblast cultures as a viable platform for SSc drug research [26]. Another use of the fibroblast cell cultures in ACTD is in the development and testing of new therapies, such as Wu and colleagues’ research in which they used skin biopsies from patients with SSc in order to limit SSc pathogenesis through fibroblast store-operated Ca^2+^ entry (SOCE) induced dedifferentiation [28]. If interested in this area, the reader is referred to more detailed reviews [9,29].

## 3. Synovial Fibroblast Cultures

Synovial fibroblasts (SF) are a type of cell found only in the stroma of the joint synovium and are its main component. Based on their localization in the synovium and their expression patterns, the cells can be distinguished into two groups, intimal and subintimal [30]. The term “synovial fibroblasts” is a broad description that refers to both subpopulations; therefore, the intimal cells are also called type B synoviocytes or fibroblast-like-synoviocytes (FLS) [31]. For the purposes of this work, both groups of cells, despite distinct differences between them, will be called synovial fibroblasts, due to the fact that both subtypes are involved in the course of RA. Apart from being the basis of the stroma’s structure, synoviocytes’ main functions are producing components of the synovial fluid and immunosurveillance [32]. Production of synovial fluid is crucial for the integrity of the cartilaginous tissue present in the joint, as well as its lubrication and mobility [33]. SF play a key role in the development and progression of RA [34] and osteoarthritis (OA) [35], but, similarly to the chondrocytes which are found in the vicinity of SF, they are also indirectly affected in the course of other ACTDs that affect the joints with their symptoms, such as JIA, spondyloarthropathies and MCTD [5,10,36]. At this point, it is important to note that OA is not considered an ACTD as it does not have an autoimmune or inflammatory background. However, OA is often investigated in ACTD research, whether as a control group in inflammatory joint damage research, in studies on joint tissue renewal therapies or as a starting point for research into the development of joint models essential for ACTDs with joint pathogenesis. In RA, due to chronic inflammation, synoviocytes undergo a series of morphological and physiological changes that result in acquiring a form of a constantly activated tumor-like RA-FLS. These cells, resistant to receptor mediated apoptosis and capable of invading other cells, are causing further inflammation and degradation of joints by the production of inflammatory factors, stimulation and the accumulation of immune cells [34]. In OA, SF, mostly FLS, begin to intensively proliferate and differentiate into myofibroblast-like cells which synthesize immense amounts of rich in fiber ECM. This process leads to accumulation of ECM and joints fibrosis causing stiffness and chronic pain [37].

The main sources of SF are aspirates of synovial fluid from joints [38], fragments of tissue collected during joint surgeries [39] and tissue harvested during arthroscopic synovectomy [40]. SF culture from synovial fluid is conducted under the standard conditions using DMEM [38]. Deriving a cell culture from tissue fragments, either harvested during surgeries or synovectomy, first requires a manual fragmentation and digestion with collagenase in DMEM for 2 h at 37 °C. When freed from the fibrous scaffold, tissues can be harvested by centrifuging [38] or filtering through a mesh [39] and cultured in DMEM with 10% FBS and 1% antibiotic–antimycotic addition [38,39,40]. Synovial fibroblasts play an important role in the pathogenesis of various ACTDs, such as RA; therefore, their cultures are a valuable tool for studying the pathogenesis of these diseases. Wei et al. performed an experiment in which they used SF collected from RA patients and revealed that the upregulation of the neurogenic locus notch homolog protein 3 (NOTCH3) signaling pathway in SF plays a significant role in inflammation and tissue degradation processes during the course of RA [41]. Moreover, SF cultures can aid in identifying novel therapy targets and testing new potential therapeutics. Three different research groups conducted experiments in which they used SF cultures derived from cells collected from RA patients has shown significant upregulation of fucosyltransferase 1 [42] and Yes-associated protein (YAP) [43] expression, as well as a shift in glucose metabolism towards glycolysis [44] suggesting them as novel targets in RA treatment strategies.

## 4. Chondrocyte Cultures

Chondrocytes are the only type of cell present in human cartilaginous tissue. They can be considered as a highly differentiated form of fibroblasts therefore their main function is synthesis of an ECM rich in proteoglycan and type II collagen [45]. Moreover, chondrocytes maintain homeostasis in cartilage by alternating between anabolic and catabolic processes of creation and degradation of ECM proteins [45]. The matrix’s functions vary based on its location within tissue or organ and the number of structural proteins it consists of. Nevertheless, its fundamental roles are supporting the growth of cells and tissues, modulation of intercellular interaction and separation of tissues. Additionally, ECM is able to alter cell behavior and growth by secretion of various factors which crucial in cellular growth, fibrosis and thereby wound healing. It has been proven that ECM plays an important role in cell differentiation, migration and even in gene expression [46,47,48].

Chondrocytes are primarily affected in OA, which causes both cells and ECM to degrade, leading to the breakdown of joint cartilage and bone structure underneath [49]. Along with the progression of the disease occurs chronic inflammation and tightening of space in the afflicted joint, due to the lack of cartilaginous tissue and growth of pathological bone structure, causing stiffness, swelling and pain [50]. Chondrocytes are also indirectly affected in the course of other ACTD such as RA, JIA, MCTD and systemic lupus erythematosus (SLE) due to the persistent, prolonged inflammation that causes joint stiffness, swelling and pain, which may lead to ECM disbalance and degradation of chondrocytes [51,52,53,54].

Chondrocyte cell cultures are widely used in OA research, both in its etiology and treatment. In the case of ACTD studies, OA is used as a control model due to the lack of an autoimmune/inflammatory background [55]. They can be also used in treatment research by testing cellular response to drugs [56,57,58] inhibiting the diseases progress using genetic engineering [59,60,61] or by coculturing chondrocytes with other cells to enhance therapy effectiveness in articular chondrocyte implantation [62].

The main source of human cartilage cells is total knee arthroplasties (TKAs Partial knee arthroplasty is an equally valid source of cells acquisition [63]. Chondrocytes can be isolated from the afore-mentioned harvested tissue and be used in setting up a primary chondrocyte cell culture in standard conditions using DMEM medium. In order to do so, cartilage must be minced or cut into very small pieces and treated with type II 0.02% collagenase at 37 °C for 4 h [64,65]. The concentration of collagenase may be lower, but the digestion time should be adequately prolonged [45,66]. However, while being cultured in 2D in vitro conditions, chondrocytes undergo a process of dedifferentiation altering their appearance into one resembling a fibroblast. Recently it has been claimed that this change also affects the expression levels of various genes and proteins, making chondrocyte cultures a less reliable tool for research [67]. Chondrocyte cultures can be used to study the etiology of RA, as described in an experiment by Andreas et al., in which they created a model consisting of postmortem-acquired human chondrocytes, supernatants from cultures of RA SF and alginate. By stimulating chondrocytes with RA SF culture supernatants, they mimicked the RA joint environment, which allowed them to identify the key regulatory molecules driving cartilage destruction [68]. Chondrocyte cultures also allow for the study of changes in cell physiology during the course of the disease. By constructing coculture models consisting of either three [69] or two [70] types of cells deriving from patients or animals, respectively, researchers were able to recreate the joint environment, which allowed the first group to establish a complete model for cartilage destruction [69] and the second group to create a perfusion culture system [70]. Chondrocyte cell cultures can also be used to observe the effects of illness on gene expression levels and gene involvement in RA pathogenesis, as described by Barksby et al. In their experiment, the researchers cultured two types of cells, chondrocytes acquired from patients and commercially acquired human chondrosarcoma cells; both were cultured in standard conditions with use of serum-free DMEM. These cells were stimulated with a combination of interleukin 1 (IL-1) and oncostatin M (OSM) to mimic cytokine environment of RA synovial fluid. Subsequently, researchers isolated RNA from the cells and performed micro-array analysis, as well as a real-time PCR, which showed an overexpression of multiple genes, such as matrix metalloproteinase 1 (MMP1), a disintegrin and metalloproteinase domain-containing protein 10 (ADAM10) and pentraxin-related protein 3 (PTX3) [71].

## 5. Peripheral Blood Mononuclear Cells

Peripheral blood mononuclear cells, also known as PBMC, is a collective term for every cell found in venous blood that has a single, round nucleus. Therefore, the main components of PBMC are lymphocytes and monocytes, with a small proportion of dendritic cells [72]. Although none of the mentioned ACTDs directly target PBMC, significant changes in the abundance of the individual cell subtypes are observed during the course of ACTDs. This leads to immunological disbalance in its subtypes, which is the main cause for prolonged inflammation and associated tissue damage in the course of these diseases [73,74]. Lymphocytes T and B, also known as T cells and B cells, are found in almost every tissue of the organism, and they are divided into several subtypes with different functionalities [75,76]. During the course of ACTD, these cells are the main driving force for inflammation and tissue damage. T cells produce cytokines which can cause the overproduction of proteins in the cell, as seen in cases of JIA [5], RA [9] and SSc [11], or can cause cells death and damage to the tissue, as seen in DM [4], LE [6], MCTD [7], PM [8] and vasculitis [12]. B cell activity in ACTDs is that of the production of antibodies targeted against the host cell’s antigens and proteins, including cellular debris, which, in combination with T cells, activity results in prolonged inflammation and further damage to tissues. Normally, monocytes and macrophages play key roles in immunological homeostasis of tissues, regulating host responses to pathogens and suppressing immune responses before they damage the tissue. But in the course of ACTDs, these cells’ functions seem to be impaired, making them unable to clear cellular debris and suppress ongoing inflammation [77]. Natural killer cells (NK cells) are responsible for detecting and destroying cells that present different antigens than those found on healthy ones and control inflammatory immune responses. There is a little evidence of NK cell contribution in ACTDs, but they are suspected to perform a similar role to that of T cells, which is secretion of cytotoxic cytokines, causing cell death and damaging the tissue [78].

The acquisition of PBMC can be performed by drawing blood from patients using tubes containing anticoagulant, usually sodium heparin, sodium citrate or EDTA [79]. It is important to note that ACTD patients often develop leukopenia, whether due to the therapeutics used in treatment or the immune imbalance caused by the disease, which might make it difficult to establish a cell culture due to insufficient cell numbers [80]. Blood collected into anticoagulant-coated tubes must be first diluted with phosphate-buffered saline (PBS) or Hanks Balanced Salt Solution (HBSS), laid on a density gradient medium and centrifuged. The volumes of PBS/HBSS and density gradient medium used depend on the amount of collected blood, whereas the centrifugation time depends on the type of density gradient. After centrifugation, PBMC, visible in the form of the buffy coat, should be recovered and washed twice in the previously used medium [81]. Subsequently, cells should be counted either automatically by using hemocytometer or manually by using cell counting chamber. After counting, cells should be cultured in RPMI 1640 with 10% FBS and 1% antibiotic-antimycotic addition [82]. PBMC cultures provide a basis for long-term experiments in controlled and standardized environments at the cost of time-consuming isolation and the imbalance of transcriptional and nutritional factors that may influence the immunological responses of cells [82]. Moreover, PBMC cultures often require additives such as concanavalin A (Con A) [83] or *Astralagus* polysaccharide (APS) [84] to enhance their proliferation. PBMCs are widely used in the research of RA and SLE as well as other ACTD. In RA PBMCs, cultures are used predominantly in the effort of understanding the nature, dynamics and intensity of the immunological and inflammatory processes in which these cells are involved, such as investigation of Wang et al. into the role of Kisspeptins during the course of RA [85]. Another studied matter is the role of PBMC-synthesized miRNAs and their pathological influence. In 2019, Zhu et al. identified an overexpression of miR-99b-5p in RA patients which was responsible for promoting lymphocyte proliferation and activation, suppression of lymphocyte T apoptosis and increase in cytokine production [86]. Moreover, expression patterns and cytokine secretion dynamics during RA are constantly researched, as shown in Garcia-Arellano et al.’s 2021 paper in which researchers identified a novel role of the macrophage migration inhibitory factor (MIF) in IL-25, IL-31 and IL-33 secretion [87]. Therapy and treatment of RA is an increasingly important field of research tasked with the investigation of novel drugs as well as their potential targets. Ramirez-Perez et al. performed an experiment using one of the therapeutics called ST2825. In their experiment, they used commercially bought PBMC samples from disease-modifying antirheumatic drugs (DMARDs)-naive patients and cultured them for 48 h with a serum-free medium and 1% antibiotic addition. After 48 h, they treated the cells with a few combinations of ST2825, lipopolysaccharide (LPS) and recombinant human interleukin 1 β (rhIL-1β) for 24 and 48 h, isolated the RNA and performed bulk RNA sequencing. What they found was that ST2825 inhibits the myeloid differentiation primary response 88 (MyD88), thereby downregulating the production of proinflammatory cytokines [88]. In 2020 de Oliveira et al. conducted an experiment in which PBMCs were isolated from blood collected from patients with diagnosed rheumatoid arthritis. Cells were cultured in standard conditions using RPMI-1640 medium and treated with two different combinations of ionomycin, phorbol-myristate-acetate (PMA) and atorvastatin for 48 h. After culturing, the researchers measured cytokine levels using flow cytometry and found that atorvastatin has a significant immunomodulatory effect in patients with RA, especially in those with severe activity of the disease. The immunomodulatory properties of the compound were expressed through the reduction of IL-6, IL-10, IL-17A and tumor necrosis factor (TNF) levels [89]. PBMC cultures are also used in the research and treatment of SLE. In lupus, PBMCs were used to understand the characteristics of gene expression in SLE. Researchers found that PBMC collected from patients diagnosed with SLE and with Toll-like receptor 7 (TLR7) rs3853839 C/G polymorphism had higher TLR7 expression and signaling than healthy controls, which promoted the production of autoantibodies by newly formed transitional B cells leading to autoreactivity, antibody activation and damage to tissues [90]. A different study on expression patterns has identified an overexpression of IL-10 in IL-10 positive B cells derived from patients with diagnosed SLE and its regulatory loop E2F2-miR-17-5p which authors suggested as a potential treatment target [91]. PBMC cultures also play a key role in the search for potential biomarkers of SLE, such as non-coding circular RNA hsa_circ_0000479 [92], ATP-binding cassette subfamily B member 1 (ABCB1), Interferon Alpha Inducible Protein 27 (IFI27) and phospholipid scramblase 1 (PLSCR1) [93]. PBMC cultures are widely used in a search for new therapeutics and treatment strategies, such as ion Zhoue et al.’s 2019 study which showed the protective effect of induced apoptosis on cells from SLE patients and suggested rapamycin and other autophagy-inducing compounds as potential novel drugs [94]. Studies on differentiation are an important aspect of PBMC research but in order to carry out such analysis, it is necessary to separate one cell subtype of interest from the isolated PBMC population. This can be achieved either by using magnetic beads or cytometry cell sorting. Separation with use of magnetic beads is technologically convenient, affordable, fast and does not require the use of sophisticated equipment. On the other hand, separation with beads is sensitive to the physical properties of the buffers used, their temperature as well as the temperature of the laboratory, the uniformity of used magnetic field and is at risk of non-specific binding [95]. Cytometry cell sorting, either fluorescence-activated cell sorting (FACS) or magnetic-activated cell sorting (MACS), allows for much more accurate cell extraction, which is relatively fast, yields high numbers of cells and allows for specific positive and negative selection. As a downside, cytometry cell sorting requires highly sophisticated and expensive equipment, well-trained staff to operate it and expensive chemicals to carry out analyses [95]. Studies on individual PBMC cell subtypes allow the analysis of their expression patterns, response to drugs, response to differentiation and growth factors and their involvement during the inflammatory process [96,97]. PBMC cultures can also be used in research of other cell types through a complex process of reprogramming and differentiation. Kim et al. performed an experiment in which they used the Sendai virus to reprogram PBMC collected from SSc patients into human-induced pluripotent cell (iPSC) lines, which were subsequently differentiated into fibroblasts and keratinocytes. Those cells were used to create skin organoids on which various selective estrogen receptor modulator (SERM) class drugs were tested to induce cell dedifferentiation [98].

## 6. Endothelial Cell Cultures

Endothelial cells are specialized cells that form the inner lining of blood vessels, including arteries, veins and capillaries. These cells are found throughout the body and play a critical role in regulating blood flow and maintaining the health of the cardiovascular system [99]. Endothelial cells form a barrier between the blood and the surrounding tissue, which prevents the leakage of blood and other fluids. They are also responsible for the production and release of various signaling molecules that help regulate blood pressure and blood vessel tone. Moreover, endothelial cells are involved in the regulation of inflammation and blood clotting [100]. Due to their high responsivity to environmental changes, such as blood flow fluctuations or the presence of inflammatory molecules, endothelial cells can change their shape and function, for example by adjusting the diameter of blood vessels or by changing the permeability of the endothelial barrier [100]. Endothelial cell dysfunction, usually in the form of vasculitis, has been observed in a number of ACTDs, including RA, MCTD, SSc and SLE [101,102,103]. Vasculitis is a condition characterized by inflammation of blood vessels, which causes damage to the walls of those vessels, leading to their narrowing, weakening or even rupture. Vasculitis can affect any blood vessel in the body, including arteries, veins and capillaries [12]. Therefore, the study of endothelial cell function and behavior is important for understanding the development and progression of vasculitis in the course of ACTD, as well as for the development of new therapies to treat it. One way to study endothelial cells is by growing them in culture, which allows researchers to manipulate and observe their behavior in a controlled environment. The first step in establishing a primary cell culture of endothelial cells is to collect the tissue from which the cells will be isolated. Endothelial cells, in the form of endothelial progenitor cells (EPCs), can be isolated from various tissues such as blood vessels [104], placenta [105] and umbilical cord [106], but also blood [107], fat [108] and bone marrow [109]. Once collected, the endothelial cells need to be isolated from other cells in the tissue. This is typically performed by using enzymatic digestion with collagenase and mechanical disruption to break down the tissue and release the cells with the exception of endothelial cells derived from blood and bone marrow, in which isolation is performed with the use of magnetic beads or flow cytometry. After isolation, the endothelial cells must usually be purified using techniques such as FACS, MACS or magnetic beads to separate them from other cell types [96]. For cells originating from blood and bone marrow, isolation is also a purification step. Purified endothelial cells must then be seeded onto a cell culture dish or flask containing a nutrient-rich medium that promotes cell growth and survival [110]. Due to the numerous endothelial cell functions, there are various types of medias that can be used for their culturing, but most commonly used are Ham’s F-12/M199, Human endothelial-SFM and EGM [110,111]. Endothelial cell cultures are used to investigate the mechanisms underlying the disease and identify potential therapeutic targets [97], as well as in diagnosing certain ACTDs, such as SLE, in which the presence of antiphospholipid antibodies is associated with an increased risk of thrombosis. Endothelial cells cultured from patients with SLE can be used to measure the level of antiphospholipid antibodies in the patient’s blood, which can aid in the diagnosis and management of the disease [112]. As mentioned before, endothelial cells also play a role in the recruitment of immune cells to sites of inflammation, therefore cultured cells can be used to test the efficacy of drugs that target this process, such as anti-adhesion molecule drugs [113]. Lastly, endothelial cells produce a number of molecules, such as C-reactive protein (CRP) and serum amyloid A (SAA) that can serve as biomarkers of disease activity or severity. Cultured endothelial cells can be used to identify these biomarkers, which can aid in the diagnosis and management of ACTD [114]. Endothelial cells can also be used in a microchip cultures, a 3D culturing method mimicking a blood vessel environment, to study the mechanisms of defective angiogenesis in the course of many ACTD such as SSc [115], psoriasis and RA [116]. Kramer et al. developed a model which allows for a high-throughput study of defective angiogenesis in the course of SSc. The model consisted of commercially available microfluidic plates seeded with polymerized neutralized rat-derived type I collagen and commercially acquired HMVECs. Cells were seeded onto polymerized collagen and treated with angiogenic factors VEGF, sphingosine-1-phospate (S1P) and PMA in EBM2 medium for 4 days to stimulate sprout formation. After stimulation, the medium was changed to standard basal medium and cells were treated with 10 ng/mL of tumor growth factor β II (TGFβ II) and tissue necrosis factor α (TNFα) in order to recreate a characteristic SSc cytokine environment [115]. Additionally, researchers studied the sprout formation with added pro-inflammatory and pro-fibrotic cytokine inhibitors and successfully showed that the addition of these inhibitors prevents cells from presenting with the diseased phenotype [115]. As Meyer and Lam state in their 2021 review on vascularized microfluidic systems, even though this technic of culturing is widely used in other research fields, such as pharmacokinetics or oncology, it is still uncommon in the research field of autoimmune diseases [116]. Microchip cultures could also be used to identify biomarkers of disease activity or severity in ACTD, to model complex interactions between immune cells and the endothelium in ACTD and, since microchip endothelial cell cultures can be generated from individual patients, they could allow for personalized treatment plans based on the patient’s specific disease and genetics [117].

## 7. Mesenchymal Stem Cells

Mesenchymal stem cells or mesenchymal stromal cells (MSCs) are a type of adult multipotent stromal cells. These cells are capable of differentiation into several specialized types of cells including myocytes, osteocytes, adipocytes, chondrocytes and neurocytes, as well as structural, connective tissue cells for stroma and tendons [118]. The main function of MSCs is maintaining a stable cell population and self-renewal through the differentiation, immunomodulation, synthesis and secretion of various growth factors and cytokines in order to maintain homeostasis and antimicrobial surveillance [118]. Even though MSCs are not directly targeted in any of the ACTDs, nor are they causative for them, they have been extensively researched and used in treatment of ACTDs [119]. MSCs’ immunomodulatory properties, such as the inhibition of T cell and B cell proliferation, increase in the proliferation of the regulatory T cells (T_regs_), the synthesis of adhesion molecules, various prostaglandins and cytokines, are used in treatment of ACTD patients in order to break the inflammation cycle and/or mitigate inflammation [119].

The main source of MSCs is bone marrow, but due to the difficulty of the collection procedure and its severity for the donor, new methods of acquisition have been sought [120]. Thus, MSCs can also be harvested from umbilical blood [121], umbilical cords [122], Wharton’s jelly [123], amniotic fluid [124], placenta [125], dental tissue [126], salivary glands [127], synovial fluid [128], body fat tissue [129] and skin [130]. Harvesting procedures can be divided into two groups, explant cultures and enzymatic cultures, although this division is artificial as isolation protocols differ from each other. In explant cultures, small pieces of cleaned and cut tissue are placed inside culture vessels containing a medium. Cells will migrate from explants into the medium, and after a few days tissue fragments can be removed [131]. Enzymatic cultures are very similar to explant ones but with an additional step of enzymatic digestion, usually with use of collagenase, hyaluronidase or dispase, between cutting and culturing in medium [131]. As mentioned before, isolation protocols are different for each MSC source, although there are several common denominators such as density gradient centrifugation, enzymatic digestion in presence of collagenase and use of DMEM, α-MEM or DMEM-F12 with a 10–20% FBS addition [131]. MSCs are widely used in regenerative therapy in patients with OA and RA, being injected into the joints affected by the disease in order to alleviate the illness and repair damaged tissue. However, due to the immunosuppressive properties of allogenic MSC, such injections may increase carcinogenic risk and lead to abnormal growth and tumor formation [132]. In order to minimize this risk, instead of MSC, injected are released by MSC extracellular vesicles (EVs) which can be divided into microvesicles, exosomes and antiapoptotic bodies [132]. Treatment with these EVs is proved to be more stable, safe and effective than with MCS [132]. MSC and MSC-EV injections are also used in slowing the progression of SSc, where their immunosuppressive and trophic properties are used to decrease skin thickness and collagen content [133,134]. Rozier et al. performed an experiment in which they used MSC-EVs to treat induced SSc in a murine model through HOCl injections. MSC-EVs were isolated from human adipose tissue cultures or from murine MSC cultures and transfected with miR-29a. Mice were injected with modified MSC-EVs and after 21 days the results were collected. The researchers showed that disease progression was slower in the mice injected with miR-29a-modified MSC-EVs and that they had an effect on the multiple metabolic pathways dysregulated by SSc [134]. Even though effects of these therapies are disputed, there is undeniable evidence confirming the positive impact on patients [133,134,135]. In OA, the current approach focuses on the improvement of regenerative treatment and the repairing properties of MSC [136,137,138], but also on modulation of the signaling pathways in order to attenuate the disease [139,140,141]. In RA, the research approach is similar, focused on finding new applications of MSC-derived exosomes [142,143,144,145,146] and new treatment procedures [144]. In SLE treatment MSC and derived from them EVs are used mostly for their wide immunomodulatory properties [145,146], but also to assess the influence of various therapeutics on those properties in order to create a combination therapies [147,148,149].

## 8. 3D Cell Cultures

All of the above-described types of cell cultures, except for endothelial microchips, are two-dimensional or monolayer cultures. Even though they have many advantages, such as relatively low cost, easy maintenance and reproducibility, they also have disadvantages [150]. The main disadvantage is the lack of cell–cell communication, which controls cell shape, holds the tissue together, prevents loss of water and solutes and gives shape, firmness, strength and durability to tissues and organs, as well as enabling the cell–cell signaling [151]. Due to the culture form, cells tend to change their morphology by flattening themselves and adhering to the base of culture vessels. This process alters not only their appearance but also their functions [150]. Moreover, cells in 2D cultures respond differently to drugs and other compounds than those in natural conditions due to the lack of an ECM and the lack of contact with other cells, as well as the exposition of every cell in the culture to the tested substance, which does not take place in reality [150]. There is also a matter of different expression levels between cells cultured with 2D methods and the same cells cultured with 3D methods [152]. Currently the best solution to this problem is the application of three-dimensional cell cultures which are able to mimic the structures and microenvironment of individual organs, therefore preserving the cells’ functions.

3D cell cultures can be divided into two types based on the technique used. In cultures using scaffolds or that are scaffold-based, cells are implanted in a synthetic substance, usually some form of hydrogel, that resembles an extra cellular matrix and allows cells to aggregate into spheroids [153]. The second type are cultures which do not use a base to form spheroids, but rather prevent the cells’ adhesion to the culture vessels, forcing them to form spheroids. The most common and simple of methods of this type is the use of U-shape bottom plates, which are constructed to prevent cells from adhering to the culture dish [154]. These base-free cultures can also be divided into subtypes based on the technique used. The forced floating technique utilizes culture vessels coated with non-adhesive agents which inhibit the cells from binding to the surfaces and enables the forming of spheroids [150,155]. Another method is the hanging drop technique, which involves placing a small drop of cell suspension in a culture dish and turning it upside down. Under the influence of gravity, the cells in the applied drop move to its bottom, where they form spheroids [150,153]. Another technique of creating spheroids is levitation, which has two variants, magnetic and acoustic. In the magnetic variant, cells are incubated and cultured with magnetic nanoparticles. Once the cells incorporate into the particles, a magnet is placed on top of the culture vessel. In the presence of a magnetic field, the cells begin to form spheroids [156]. The acoustic variant achieves the same result, but instead of using nanoparticles and magnets, it utilizes ultrasonic resonators which cause cells to hover in the culture medium [153]. Spheroids can also be cultured with the use of motion, which can be achieved by shaking, rotating or stirring. Constant movement prevents cells from adhering and forces them to take the structure of spheroids [157].

Next method is bioprinting, which is really 3D printing downscaled to a microscopic scale. This means it is able to create biological structures by applying layer after layer of cells to reflect the natural form of these structures. While previous methods generated spheroids, bioprinting allows for the creation of whole tissues depending on the bioinks used [157]. The last covered method is microfluidic systems which allow for the formation of spheroids through the culturing of cells in microscopic chambers connected by medium flowing channels [157,158].

3D cultures are predominantly used in OA and RA research, although they might also be useful tools in SLE research. In 2022, Park et al. conducted an experiment in which they used cardiomyocyte spheroids to successfully establish a model for lupus heart disease. First, researchers collected PBMC from SLE patients and transformed them into iPSCs. Then, they treated the cells with a cardiomyocyte differentiation medium to transform them into cardiomyocytes. Lastly, differentiated cells were seeded onto a 96-well U-shaped plate to form spheroids [159]. There is also ongoing research to utilize salivary gland organoids in studies on Sjögren’s syndrome (SS), during which lymphocytic infiltration of the salivary glands and mucous membranes results in their damage and dysfunction [160]. However, at the date of writing and to the authors’ knowledge, there are no publications showing the use of salivary gland organoids in ACTD research, as this type of culture is still in the developmental stage. With the use of 3D tissue and organ models, scientists are able to more accurately assess the intensity and effectiveness of the researched drugs and compounds. Buhrmann et al. designed a 3D culture model consisting of chondrocyte cells encapsulated in alginate beads, fibroblasts and Jurkat cells (human T lymphocytes) to recreate osteoarthritic joint environment. Having established the model and osteoarthritic background, scientists tested different concentrations of curcumin and found that it weakens the osteoarthritic background by regulating the SRY-Box Transcription Factor 9 (Sox9)/nuclear factor kappa-light-chain-enhancer of activated B cells (NF-kB) signaling axis [57]. Three-dimensional cell cultures also enable scientists to study the etiology of diseases by mimicking their pathological environments in vitro [161,162]. Bundens et al. also established a 3D model of osteoarthritic chondrocytes, but in their model chondrocytes were the only type of cell present. After collection from patients, cartilage was enzymatically broken down into free cells, which were suspended in alginate and formed into beads. These beads were then cultured in high-glucose DMEM. Scientists assessed the chondrocyte activity and their similarity to in vivo conditions by measuring proteoglycans, collagen production and degradation as well as cytokine production. What they found was that cultured chondrocytes maintained their proteoglycan, collagen and cytokine production, while keeping their OA characteristics [162]. With the use of similar 3D culturing methods, scientists are also able to understand expression patterns in healthy and pathological tissues that are closer to in vivo conditions [163,164]. Three-dimensional cell cultures also provide a more accurate environment for testing novel treatment therapies and finding new therapeutic targets [63,165,166]. Caire et al. utilized synovial organoids from RA patients in order to identify the greater transcriptional activity of the coactivators YAP/TAZ (where TAZ is WW-domain-containing transcription regulator 1) in rheumatoid cells in comparison to the controls, suggesting YAP/TAZ as possible target for treatment. The organoids were created by mixing synovial cells with a gel-like matrix and droplet seeding onto a 96-well U-shaped bottom low-attachment plate and culturing them for 21 days in the presence of a standard DMEM with 10% FBS and antibiotics, ascorbic acid and ITS solution [165].

## 9. Cell Culture Skin Models

Skin is the largest organ of the body and serves as a protective barrier between the external environment and the internal organs [167]. It is made up of the following three primary layers: the epidermis, dermis and subcutaneous tissue. The epidermis is the outermost layer of the skin and is composed of several layers of cells, including keratinocytes, melanocytes and Langerhans cells [167]. The keratinocytes are responsible for producing a tough, fibrous protein called keratin, which helps to provide strength and durability to the skin. Melanocytes produce melanin, the pigment that gives the skin its color, while Langerhans cells are involved in the immune response to foreign substances that come into contact with the skin [168]. The dermis is the middle layer of the skin and is composed of connective tissue, blood vessels and nerve endings. It provides support and nourishment to the epidermis and contains structures such as hair follicles, sweat glands and sebaceous glands. The dermis also contains fibroblasts, which are cells that produce collagen and elastin, two proteins that provide strength and elasticity to the skin [169]. The subcutaneous tissue is the deepest layer of the skin and is composed of adipose tissue and connective tissue. It serves as a cushioning layer that helps to protect the internal organs and assists in the regulation of body temperature [169]. The skin has many important functions, including the protection against physical, chemical and microbial damage, the maintenance of the thermal balance of the body and the synthesis of vitamin D, sensation of touch, pressure and pain. The skin also plays a role in immune surveillance, as it contains immune cells [170] that help to identify and eliminate pathogens and abnormal cells [171]. Several ACTDs, such as SSc, SLE and DM, can lead to skin fibrosis and desquamation. As mentioned before, skin fibrosis is caused by the activation and proliferation of fibroblasts, which are cells that can produce ECM proteins such as collagen. Fibroblasts become activated in response to the cytokines and growth factors produced by the immune cells, leading to excessive collagen production and deposition in the skin and other organs [14]. Desquamation of the skin refers to the shedding or peeling of the outer layers of the skin. In the course of ACTDs, desquamation of the skin can occur as a result of several factors, depending on the disease. In SLE, desquamation of the skin can be a result of cutaneous lupus erythematosus (CLE), a type of lupus that affects the skin. CLE can cause a variety of skin lesions, including scaly or flaky patches of skin that can peel or slough off. In addition, medications used to treat SLE, such as hydroxychloroquine, can cause desquamation of the skin as a side effect [6]. In DM, desquamation of the skin can occur as a result of skin inflammation and damage. DM is a rare autoimmune disease that affects the skin and muscles and can cause a range of skin changes including redness, scaling and peeling [4]. In SSc, desquamation of the skin can be a result of skin dryness and damage caused by fibrosis and vascular changes in the skin. As the skin becomes thicker and less elastic, it can become dry and prone to cracking and peeling [14].

A cell culture skin model usually refers to a three-dimensional culture system that mimics the structure and function of the human skin, although there are established 2D skin models [172]. In ACTD studies, it is used to better understand the pathogenesis of these diseases and to develop new treatments. Skin models can be either in vitro, ex vivo or in vivo [173]. The latter will not be discussed in this paper as in vivo skin models are not cell culture models, similarly to in vitro membrane models. In vitro skin models are created using cultured cells that are grown in a laboratory setting. The first step to establishing an in vitro skin model is the isolation of the cells, typically derived from human skin biopsies or commercially available cell lines [174]. The most common types of cells used in skin models are keratinocytes, which make up the epidermis and fibroblasts that make up the dermis. Once isolated, the cells are seeded onto a scaffolding or matrix that provides a three-dimensional environment for them to grow and interact. The most commonly used scaffolds are collagen, hyaluronic acid, and synthetic polymers [175]. The seeded scaffold is then placed in a culture dish or flask containing a nutrient-rich medium that promotes cell growth and survival. The medium may also contain supplements such as growth factors, cytokines and serum to enhance cell proliferation and function [176]. To mimic the complex interactions between different cell types in the skin, dermal models usually include multiple types of cells. Over time, the cells in the model will differentiate and form the different layers of the skin, such as the epidermis and dermis [177]. This process can be enhanced by modifying the culture’s conditions, such as by changing the composition of the medium or applying mechanical stress to the model. Once the skin model has formed, it can be characterized to confirm that it is indeed a functional model of human skin. This can be performed using various techniques, such as immunostaining for skin-specific markers or functional assays to measure skin behavior, such as barrier function, wound healing or inflammation [177].

The ex vivo skin models differ from the in vitro models in the way the cells are obtained and the culture established, while the maintenance of both types of culture is similar. Human ex vivo skin models are skin samples that are removed from humans and cultured in a laboratory setting. Ex vivo skin models can be created using various techniques. The most common method focuses on taking skin biopsies or skin explants from patients and culturing them in specialized medias, usually DMEM with a 10% FBS and 1% antibiotic–antimycotic addition, maintained in standard conditions which mimic the in vivo environment [178]. Such cultures retain much of the structure and function of intact skin as well as its vascularization. These cultures can be used to study the biology of the skin and to test the effects of drugs or other treatments on skin cells without the confounding effects of other organ systems or the immune system. Explant-derived skin models can also be used to study the cellular and molecular mechanisms of skin-related ACTDs [177]. Aden et al. conducted research on skin biopsies obtained from patients with SSc, in which they investigated the role of epidermal cells in the fibroblast activation in SSc. Firstly, fibroblasts were seeded in 24-well plates with mixture of standard DMEM and collagen type I. Then, epidermal sheets, created through enzymatic digestion of skin biopsies, were added to the wells containing fibroblasts and were cocultured together. The researchers measured levels of proinflammatory cytokines and found that epidermal cells promoted fibroblast activation through IL-1α, as well as the stress response signaling pathways being induced in SSc epidermal cells [179]. Another way of generating skin models is the coculture of different types of cells of interest. Using SSc as an example, commercially bought fibroblasts and keratinocytes, after thawing and expanding, can be seeded together and maintained in standard conditions with standard addition of DMEM. Through the addition of TGFβ, a known fibrosis factor in SSc, it is possible to induce pathological responses in cells similar to those observed in vivo, therefore creating a SSc model that allows for the measurement of protein production, gene expression or drug response [177].

## 10. Summary

Overall, cell cultures can serve as a valuable tool in the research and treatment of autoimmune connective tissue diseases. They provide insights into disease mechanisms, enable biomarker discovery, facilitate drug development and screening and, in the near future, they could support personalized medicine approaches and contribute to the development of patient-specific therapies. Through the use and development of cell cultures in ACTD research and treatment, scientists and clinicians can advance our understanding of these diseases and improve patient outcomes.

## Figures and Tables

**Table 1 cells-12-02489-t001:** List of described ACTDs, tissues they affect, type of cell cultures that might be used in research on particular diseases, immune elements involved in pathological processes and their immunopathogenesis.

ACTD	Affected Tissue	Type of Cell Culture	Involved Immune Elements	Immunopathogenesis
Dermatomyositis (DM)	Muscles, skin [4].	2D and 3D fibroblast cultures and skin models.	Myositis-specific antibodies: nuclear matrix protein 2 (anti-NXP-2), transcriptional intermediary factor 1 γ (anti-TIF1-γ), melanoma differentiation-associated protein 5 (anti-MDA-5), small ubiquitin-like modifier activating enzyme 1 (anti-SAE-1), subunit of nucleosome remodeling deacetylase complex (anti-Mi-2) [4].	Characterized by a chronic inflammatory process affecting the skin and muscles. Infiltration of the affected tissues by immune elements leads to tissue damage and inflammation. Also associated with microangiopathy in affected tissues, which contributes to skin changes, muscle weakness and other clinical features [4].
Juvenile idiopathic arthritis (JIA)	Joints, cartilage, bones [5].	2D and 3D fibroblast and chondrocyte cultures.	Macrophages and T lymphocytes [5].	Overactive immune response, characterized by an influx of T cells and macrophages into the synovium and the release of proinflammatory cytokines, which promotes inflammation and tissue damage in the joints. Various subtypes with significant differences in pathogenesis [5].
Lupus erythematosus (LE)	Various, including joints, blood cells, brain, heart, skin, kidneys, lungs [6].	2D and 3D fibroblast, chondrocyte and cardiomyocyte cultures, peripheral blood mononuclear cells (PBMC) cultures, skin models.	Defective autoantibodies recruiting T and B lymphocytes [6].	Dysregulated immune response, leading to the activation of various components of the immune system. Imbalance between proinflammatory and regulatory immune responses. Production of autoantibodies which target and attack the body’s own tissues. Formation of immune complexes by antibodies with their target antigens, complexes can deposit in various tissues leading to inflammation and tissue damage [6].
Mixed connective tissue disease (MCTD)	Various, including skin, muscles, small capillary vessels, lungs, kidneys [7].	2D and 3D fibroblast, chondrocyte and endothelial cell cultures, PBMC cultures.	Anti-U1-RNP antibodies [7].	Immunological imbalance features similar and overlapping with lupus, systemic sclerosis and polymyositis [7].
Polymyositis (PM)	Muscles [8].	2D and 3D fibroblast and myocyte cultures.	CD8+ T cells [8].	Infiltration of CD8+ T cells into the muscle tissue, which recognize and attack muscle fibers, leading to muscle inflammation and damage. Persistent inflammation interferes with the muscle regeneration processes. As muscle fibers are damaged and replaced by fibrous tissue, muscle strength and function are compromised [8].
Rheumatoid arthritis (RA)	Joints, cartilage and bones [9].	2D and 3D fibroblast and chondrocyte cultures.	CD4+ T helper cells, rheumatoid factor (RF), anti–citrullinated protein antibodies (ACPA) [9].	Dysregulation of immune response with activation of CD4+ T helper cells and production of proinflammatory cytokines. Synthesis of RF and ACPA autoantibodies, leading to inflammation and tissue damage, mostly in joint areas. Formation and deposition of immune complexes in tissue which contribute to inflammation. Cytokine-stimulated bone erosion and deformation caused by osteoclasts [9].
Spondyloarthropathies	Joints of the vertebral column [10].	2D and 3D fibroblast and chondrocyte cultures.	Human leukocyte antigen B27 (HLA-B27) antibodies [10].	Inappropriate immune response, characterized by the activation of immune cells, leading to chronic inflammation and infiltration of the affected joints, including the spine and peripheral joints. Overproduction of proinflammatory cytokines contributing to enthesitis and tissue damage. Various subtypes with significant differences in pathogenesis [10].
Systemic sclerosis (SSc)	Skin, soft tissue organs, blood vessels, joints [11].	2D and 3D fibroblast, PBMC, mesenchymal stem cell (MSC) and endothelial cell cultures, skin models.	T cells, CD4+ T cells [11].	Persistent activation of fibroblasts, leading to imbalance in extra cellular matrix (ECM) production and degradation processes resulting in fibrosis. Activation of endothelial cells leads to an abnormal release of vasoactive factors, resulting in blood flow changes and vessel dysfunction. Infiltration of the affected tissues by inflammatory cells and release of proinflammatory cytokines and chemokines result in tissue damage [11].
Vasculitis	Various blood vessels [12].	2D and 3D endothelial cell cultures, PBMC cultures.	Anti-neutrophil cytoplasmic antibodies (ANCAs) [12].	Dysregulation of the immune system, leading to the activation of immune cells and production of proinflammatory cytokines and chemokines resulting in damage to vessels. Activation of endothelial cells, resulting in the promotion of immune cell infiltration. Various subtypes with significant differences in pathogenesis [12].

**Table 2 cells-12-02489-t002:** List of medias used in the described cell cultures with their composition and application.

Medium Name	Composition	Application
Dulbecco’s Modified Eagle Medium (DMEM)	Glucose, amino acids, vitamins, minerals and a buffering agent to maintain a constant pH. Available in numerous modifications, including increased glucose concentration, sodium pyruvate addition or prolonged shelf life [15].	A wide range of cell types, including fibroblasts, epithelial cells, smooth muscle cells, neurons, glial cells and certain cell lines [15].
Ham’s F-12	Modification of Eagle Medium similar in composition to DMEM but with a higher concentration of bicarbonate, which helps to regulate the pH of the medium and maintain proper osmotic pressure [15].	A wide range of cell types, including epithelial cells, fibroblasts, neurons, endothelial cells, hepatocytes, adipocytes and cells that are sensitive to changes in osmotic pressure [15].
M199	Modification of Eagle Medium similar in composition to DMEM but with high concentration of inorganic salts, which help to maintain the proper osmotic pressure and provide a stable environment for the cells [16].	Fibroblasts, endothelial cells, epithelial cells, hybridoma cells, hepatocytes, smooth muscle cells and cells sensitive to osmotic pressure changes [16].
α-Modified Eagle Medium (α-MEM)	A modification of the original minimum essential medium (MEM) with high concentration of vitamins and low concentration of pyruvate [16].	Fibroblasts, epithelial cells, stem cells, hybridoma cells, adipocytes and osteoblasts [16].
Roswell Park Memorial Institute (RPMI) 1640	A modification of MEM supplemented with additional compounds, including HEPES (4-(2-hydroxyethyl)-1-piperazineethanesulfonic acid), which helps to maintain the pH of the medium in the presence of carbon dioxide and sodium bicarbonate, which helps to buffer the medium [15].	Lymphocytes, cancer cells, hybridoma cells, adipocytes, fibroblasts and epithelial cells [15].
Human Endothelial Serum Free Medium (Human Endothelial SFM)	Serum-free medium with a range of supplements, including growth factors, amino acids, vitamins and trace elements that are essential for EC growth, such as Vascular Endothelial Growth Factor (VEGF), Basic Fibroblast Growth Factor (bFGF) and Epidermal Growth Factor (EGF) [17].	Human ECs, such as human umbilical vein endothelial cells (HUVECs), human dermal microvascular endothelial cells (HDMECs), human pulmonary microvascular endothelial cells (HPMECs) and human coronary artery endothelial cells (HCAECs) [17].
Endothelial Cell Growth Medium (EGM)	Variety of nutrients, growth factors and supplements that are necessary for the growth and survival of ECs, which can include growth factors such as VEGF, bFGF and EGF, as well as other components, such as hydrocortisone, ascorbic acid and heparin [18].	It is important to note that EGM medium is formulated for specific types of ECs. Therefore, different ECs will require different EGMs. Some of the cells that can be cultured in this medium are HUVECs, human microvascular endothelial cells (HMVECs), human pulmonary artery endothelial cells (HPAECs), HDMECs, HCAECs and human retinal microvascular endothelial cells (HRMECs) [18].

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
