# Peer review of "Cell Cultures as a Versatile Tool in the Research and Treatment of Autoimmune Connective Tissue Diseases"

_cells, 2023, doi:10.3390/cells12202489_

Round 1

Reviewer 1 Report

The authors present a review of the literature on the use of different types of cell cultures as a research tool in connective tissue diseases. I consider the structure to be adequate with a mention of the different techniques, and the text to be clear and easy to read.

The only point to note is that the processing techniques are described for all the tissues mentioned, except fibroblasts, for which only two more detailed reviews are given. A description of the technique and the main sources of fibroblast acquisition is desirable for other cell cultures.

Otherwise, I consider this an extensive and laborious review.

Author Response

Thank you very much for your insight, I appreciate it. I added a paragraph containing brief processing procedure to the Fibroblast section and highlighted it.

Reviewer 2 Report

 General:

The submitted manuscript is the review presenting at the short and careful form the today’s cell culture methods used for the study of autoimmune connective tissue diseases in vitro. The manuscript describes approaches for cell separation from the affected tissues, cell media’ formulations, 2D and 3D cell culturing models and thus is good practical manual for the researchers and students of the scientific labs. It merits publication and would be of interest to the readership of the journal as well as to other researchers at biomedical sciences.

However, some, mostly typographical, errors need to be corrected.

Minor Concerns:

1.      Lines 131, 176. Incorrect usage the degree symbol (37oC). Please use the standard superscript form.

2.      Lines 161-162. Abbreviations should be defined when they appear first time (lines 114-117). Please correct.

3.      Line 182. Comma is absent. Please add.

Conclusion:

Manuscript should be accepted and published after the minor revision.

Author Response

Thank you for your insight and keen eye. I corrected the degree symbols notation, properly added abbrevations when they appear for the first time and added coma at the end of the sentence in line 182. I also found few additional mistakes which I corrected, every correction was highlighted.

Reviewer 3 Report

Ejma-Multanski and colleagues have conducted a comprehensive review delineating the significance of cell cultures as a valuable tool within the realm of autoimmune connective tissue diseases (ACTD) research, along with their potential therapeutic applications. While the subject matter undoubtedly garners considerable interest across diverse audiences, there are several concerns and recommendations that warrant meticulous consideration to enhance the overall clarity, depth, and impact of the findings presented.

1.       The manuscript primarily centers on the cell culture technique for investigating ACTD and the authors extensively describe how to isolate and culture the cells. Nonetheless, it is imperative to provide a foundational background of ACTD within the introduction. Not even the name of any single ACTD was mentioned in the introduction. At least, a brief overview of these diseases, such as the target tissues and immune system cells involved, should be introduced to establish context.

2.       The manuscript contains a surplus of information on topics not directly related to ACTD, specifically in sections spanning lines 69-82 and lines 187-216. To enhance precision, the authors should briefly delineate the characteristics and functions of cells and directly focus more explicitly on their roles in ACTD development rather than making general statements about their involvement in various diseases such as “… also play a role in regulating immune responses and are involved in the development and progression of various diseases, including ACTD”

3.       One of the major concerns for the manuscript is that while the isolation and culture of cells are expounded upon, the manuscript is deficient in elucidating how these cultures are applied to recapitulate or study ACTD and merely touches upon the utility of these cultures in understanding the underlying mechanisms of ACTD pathogenesis. For example, in lines 96-101 for fibroblast cultures, the author just said superficially that “One use of fibroblast cultures in aforementioned diseases is to study the underlying mechanisms of the pathological processes involved in their development and progression” or “Synovial fibroblasts play an important role in the pathogenesis of various ACTD such as RA, therefore their cultures are a valuable tool for studying the pathogenesis of these diseases.” in lines 134-136 for synovial fibroblast cultures or “Chondrocyte cultures can be used to study the etiology of RA and changes in cell physiology during the course of the disease” in lines 182-183 for chondrocyte culture. But how??? These cells were cultured for various types of diseases, but how we can use them specifically for studying or modeling ACTD? Or the same cells like fibroblasts play a role in several ACTD such as SSc, DM and PM (lines 86-87), how the same protocol for isolating and culturing fibroblasts could be used for studying different ACTD?  The authors should expound on the practical applications of these cell cultures in understanding the pathogenesis of ACTD, offering concrete examples and methods.

4.       Same thing regarding other parts such as for 3D cultures. The authors just stated “3D cultures are predominantly used in OA and RA research, although they also 432 might be useful tool in SLE research”, however, no further explanation on how to apply these 3D cultures in ACTD research. It is crucial to detail how 3D cultures can be effectively employed in the context of ACTD.

5.       The table and Figure included in the manuscript are not obviously helpful in elucidating the topic. Table 1 is very technically oriented, too detailed but not really relevant to the topic, particularly for the application listed in the table could not show how to apply this media for ACTD research (some of the cell types shown in the application on the table were not clearly involved in ACTD). Figure 1 solely depicts types of peripheral blood mononuclear cells (PBMCs) without providing substantial insight into cell culture technique or ACTD at all. The authors might consider incorporating other tables/figures that are directly related to ACTD such as a table summarizing a list of ACTD, the cells that are involved in its pathogenesis, and how to use cell culture to study it or a figure showing different methods to culture the cells in ACTD research (2D vs different types of 3D cultures, microchips)

6.       The PBMCs section requires refocusing to emphasize their relevance to ACTD. The current presentation lacks sufficient emphasis on ACTD, particularly in lines 195-216, which predominantly explains the functions of each cell type in PBMCs in general, rather than their role in ACTD. Furthermore, again, there was no detail on how to culture these PBMC cells for ACTD and it is still difficult to imagine how these PBMCs culture as surrogate cells to study ACTD? Lines 246-249 regarding ST2825 therapy, have no explanation of how the PBMC culture was performed and help to investigate novel drugs. A similar thing happened to lines 250-253 regarding atorvastatin on RA. Line 254-260, I believe that the studies cited here (ref. #86, 87) did not use any PBMC culture at all. Both studies freshly isolated PBMCs and used them directly without culture?

7.       Lines 331-335, it is worth illustrating and emphasizing more on how the 3D-cultured microchip or organ-on-chip were created so the audience can have a picture of how they look like and how they can mimic the ACTD. Were they used endothelial cells isolated from patients? However, it is unclear to me why the authors mention these 3D cultures in the “endothelial cell cultures” section but did not just combine this part into the “3D cell cultures” section.

8.       Lines 351 -354, The manuscript should first elaborate on how mesenchymal stem cell cultures can be applied in ACTD therapy prior to detailing how to isolate and culture these cells. Also, not sure what the authors mean in lines 372-374: “Due to the possible risk of abnormal growth and tumor formation instead of MSC are injected derived from them extracellular vesicles (EVs) which are more stable, safe and effective than their progenitors”?

9.       Not clear what is special about skin over other organs, so the authors discussed it separately as another “skin model” section. Similar to the above-mentioned issues in other sections, the authors comprehensively discussed how to generate skin models and their applications in general, but the authors failed to show how to mimic ACTD diseases on these skin models and how to use them for studying ACTD specifically.

10.   The abstract should be reorganized to align more coherently with the manuscript's content. Not sure why only fibroblast cultures and peripheral blood mononuclear cell cultures were mentioned in the abstract, while several other cell types and 3D models were discussed throughout the manuscript.

11.   Several redundant statements that might consider merging such as lines 73-76: two sentences were talking similarly.

12.   Please ensure that all abbreviations are properly introduced with their full names upon first use in the manuscript such as MDA-5 (line 94), SOCE (line 99), SERM (line 100), RA (line 110), MCTD and JIA (line 117), etc.

Author Response

List of authors' changes according to reviewer's remarks, all changes are highlighted yellow.

1. Introduction was altered to present more information on mentioned ACTD through a table 1.

2. Sections mentioned by the reviewer were shortened and now describe cells' impact in the course of ACTD.

3. Paragraph describes more precisely use of such cultures, practical examples were expanded upon.

4. Similarly to point 3, paragraph now decribes use of such cultures in ACTD more precisely and with examples.

5. Table of ACTD mentioned in the paper, as well as cells they involve and immune cells was added to Introduction paragraph.

6. PBMC section is now more precise, describes PBMC roles in ACTD, use of cultures is written more precisely with more examples.

7. The organ on a chip model was described more precisely, also examples were expanded upon.

8. Application of MSC in ACTD research was restructured to be before isolation protocol, the unclear line on 372 - 374 was rephrased.

9. Similarly to paragraphs before, paragraph describes more precisely use of such cultures, practical examples were expanded upon.

10. Abstract was reorganized, every cell culture model present in the paper is now mentioned in it.

11. Redundant statements were merged together.

12. Abbreviations are now properly introduced.

Round 2

Reviewer 3 Report

I appreciate the thoughtful response from the authors, the majority of which effectively addressed the raised concerns. Although the manuscript is substantially improved, there remain some specific issues that require attention:

1.       I commend the authors for the inclusion of Table 1. However, it would be more informative if the authors provided a brief description of the immunopathogenesis, in addition to listing the involved immune cells (and actually, the authors might consider changing the title of that column to be broader terms since the author listed some of the non-cell contributors such as antibodies which are not cells) And to align with line 47 it may be beneficial to include information about the cell culture type that could be advantageous for studying that particular ACTD. (I acknowledge that I did not explicitly state this in my previous review.)

2.       Line 109-111, it appears that the authors should reorganize the discussion to focus on fibroblasts. It is unclear why the authors are discussing SSc skin biopsies in the "fibroblast cultures" section. And instead of discussing “SSc expression profiles did not change whether they were collected from affected area or not” (which it is not clear how would this help to emphasize the benefits of using fibroblast cultures to study SSc), from ref #26, the key point of the cited study should be “SSc dermal fibroblasts retained most of the molecular disease phenotype upon in vitro culture” and “results demonstrated the value of carefully-phenotyped SSc dermal fibroblasts as a platform for SSc target and drug discovery.”

3.       Line 115-119, it seems more appropriate to relocate this section to the "PBMCs" section, as it demonstrates the utility of PBMCs for reprogramming into other cell types.

4.       One notable concern I have, which may have been overlooked in the original review, is the recurring mention of OA several times throughout the manuscript as an autoimmune disease [such as lines 133, 142, 177, 187, 444, 513-526]. It is important to clarify that OA and RA are distinct conditions with different underlying causes. RA is an autoimmune condition whereas OA seems not. Indeed, the authors themselves acknowledged in line 187 that OA lacks an autoimmune or inflammatory background. Consequently, the appropriateness of using OA as an example of a cell culture model for studying ACTDs is questionable. If this choice is retained, the manuscript should explicitly state the rationale for including OA despite it not being considered one of the ACTDs.

5.       The utility of Figure 1 in this manuscript remains unclear. It does not effectively illustrate how PBMCs can contribute to our understanding of ACTDs. Furthermore, questions arise regarding the accuracy of the drawings in terms of relative size and the morphology of cells and nuclei. And I doubt how many macrophages (look like shown in Fig. 1F) could be found in PBMCs?

6.       Line 323-330, the terms of magnetic vs FACS sorting should be mentioned. So might consider moving Line 362-365 up before discussing them.

7.       Line 446-451, the description of applications of MSC-derived exosomes in ACTD treatments appears somewhat superficial. This section should be expanded to provide a more comprehensive understanding of the subject matter.

8.       Line 72, not sure what “so much” mean

9.       Line 272, please italicize “Astragalus

10.   Line 218, 291, 310, please lowercase “interleukin” and indeed the authors need to state the full name as interleukin (IL) just for the first time in line 218 (do not need it again in line 284 as well)

11.   Line 306, please confirm the abbreviation of “Toll-like receptor 7”. TCF7 or TLR7?

12.   Line 475, please delete “-” in “pre-vents”

Author Response

Thank you very much for your keen and thorough insight. We have made the suggested changes in paper which we address lower:

  1. Added "Cell culture type" and "Immunopathogenesis" columns with types of cell cultures beneficial for specific diseases and brief description of their pathogenesis.
  2. Rearranged the topic to center around fibroblasts and changed the key points relevant to the reference study.
  3. Line was relocated to the end of PBMC section.
  4. Added an explanation for describing OA cultures and treatments in the field of ACTD treatment when OA appears for the first time in text.
  5. Removed the figure 1 as it's purpose is no longer relevant.
  6. Line was moved to introduce FACS earlier in text.
  7. Description of usage of MSC-Evs in ACTD treatment was expanded with an example of such use.
  8. Term was removed and replaced with more elegant grammatical form.
  9. Astralagaus was italicized.
  10. Interleukin was lowercased and it's repetitions were shortened to IL.
  11. Abbreviation of Toll-like receptor 7 was correctly written as TLR7 (our second study secretly snuck in with TCF7)
  12. Hyphen in "pre-vents" was removed.